# Gum Arabic Increases Phagocytosis of *Escherichia coli* by Blood Leukocytes of Young and Old Healthy Volunteers

**DOI:** 10.3390/antibiotics13060482

**Published:** 2024-05-24

**Authors:** Christin Freibrodt, Shima Baien, Maren von Köckritz-Blickwede, Nicole de Buhr, Roland Nau, Jana Seele

**Affiliations:** 1Department of Neuropathology, University Medical Center Göttingen, Georg-August-University Göttingen, 37073 Göttingen, Germany; christin-freibrodt@web.de; 2Department of Geriatrics, Evangelisches Krankenhaus Göttingen-Weende, 37075 Göttingen, Germany; 3Department of Biochemistry, University of Veterinary Medicine Hannover, 30559 Hannover, Germanymaren.von.koeckritz-blickwede@tiho-hannover.de (M.v.K.-B.); nicole.de.buhr@tiho-hannover.de (N.d.B.); 4Research Center for Emerging Infections and Zoonoses (RIZ), University of Veterinary Medicine Hannover, 30559 Hannover, Germany

**Keywords:** *Acacia senegal* (L.) Willdenow tree, gum arabic, phagocytosis, bactericidal effect, TNF-α, IL-6

## Abstract

Background: Gum arabic, a polysaccharide exudate from *Acacia senegal* (L.) Willdenow trees, has already been used by African native people in natural medicine. Methods: Using whole-blood samples from young (20–35 years) and older (>80 years) healthy volunteers (each group n = 10), the effect of an aqueous solution of GA on phagocytosis of *Escherichia coli* was examined with a gentamicin protection assay. Whole-blood samples of each volunteer were stimulated with GA and as a control with CpG oligodeoxynucleotides (Toll-like receptor -9 agonists) for 2 h, then co-incubated with *E. coli* for 30 min and thereafter treated with gentamicin for up to 240 min to kill extracellular bacteria. Then, whole-blood cells were lysed with distilled water, and colony-forming units were counted by quantitative plating. Cytokine enzyme-linked immunosorbent assay for the detection of TNF-α and IL-6 was performed using the blood supernatant. Results: The GA concentration tested (20 mg/mL) did not affect the viability of eukaryotic cells. Phagocytosis of *E. coli* by whole-blood leukocytes derived from young (*p* = 0.008) and older (*p* = 0.004) healthy volunteers was increased by 120.8% (young) and 39.2% (old) after stimulation with GA. In contrast, CpG only stimulated the bacterial phagocytosis by cells derived from young volunteers (*p* = 0.004). Stimulation of whole blood with GA increased the intracellular killing of *E. coli* in young (*p* = 0.045) and older volunteers (*p* = 0.008) and induced a TNF-α release in whole blood collected from older volunteers but not from younger ones (*p* = 0.008). Conclusions: These data encourage the isolation of active compounds of GA and the initiation of clinical trials addressing the preventive effect of GA on bacterial infections.

## 1. Introduction

Antimicrobial resistance is an increasing threat causing approx. 700,000 deaths each year worldwide [1]. For the antibiotic treatment of some bacterial strains, almost no options remain. The conditions are more dramatic with Gram-negative organisms, e.g., carbapenem-resistant *Escherichia coli* and other enterobacteria, *Acinetobacter baumannii* and *Pseudomonas aeruginosa* strains, than with Gram-positive bacteria, e.g., methicillin-resistant staphylococci, vancomycin-resistant enterococci and multidrug-resistant *Streptococcus pneumoniae*. The greatest therapeutic challenges are caused by infections with Gram-negative bacteria extensively drug-resistant (XDR, resistant to ß-lactam antibiotics, fluoroquinolones, aminoglycosides and colistin) or co-resistant to carbapenems, aminoglycosides, polymyxins and tigecycline (CAPT-resistant) [2,3].

For the treatment of infections, where effective treatment with licensed antibiotics is not available, immunomodulatory or antibacterial natural substances appear promising. For this reason, an extensive search has started to identify natural products with antibacterial or immunomodulatory properties. Gum Arabic (GA), a dried gummy exudate from stems and branches of the *Acacia senegal* (L.) Willdenow [synonym: *Senegalia senegal* (L.) Britton] tree, is a water-soluble polysaccharide with sugars including rhamnose, arabinose, and galactose also containing highly branched complex arabinogalactan proteins, glucuronic acid and minerals [4]. In the European Union, it is listed as a food additive [5]. It has been used in traditional African medicine to treat a variety of infections. At a daily oral dose of approximately 10 g, it acted as a probiotic, increasing the numbers of *Bifidobacteria*, *Lactobacilli* and *Bacteroides* spp. in the stools of healthy volunteers [6]. In vitro, GA exhibited a direct antibacterial effect against some *Staphylococcus aureus* and *E. coli* strains, and increased the phagocytosis of *S. aureus* bioparticles by bovine granulocytes [7]. In blood derived from young healthy volunteers, GA stimulated phagocytosis and intracellular killing of an encapsulated *E. coli* strain [7].

Aging of the immune system is a physiological process affecting both the innate and the adaptive immune systems. The immune system of older humans is characterized by a basal activation (e.g., a slight increase in C-reactive protein and elevated concentrations of pro-inflammatory cytokines in plasma) coupled to a reduced ability to adequately react upon infections (“*Inflamm-Aging*”) [8]. In particular, the ability of granulocytes and macrophages to phagocytose pathogens and the ability of B-lymphocytes to synthesize accurately fitting antibodies are affected. As a consequence of immune system aging, older individuals are particularly vulnerable to infections [9,10]. Since they often receive repeated antibiotic treatments and have contact with other people who are exposed to antibiotics (in nursing homes and hospitals), older persons are particularly endangered by multi-resistant bacteria.

GA has immunomodulatory properties on leukocytes prepared from young persons and has a direct antibacterial effect on some pathogens [7]. As immunomodulatory compounds which are able to increase the infection resistance of older individuals are desperately needed, this study aims at assessing the immunomodulatory effect of GA on bacterial phagocytosis and intracellular killing in blood leukocytes of older volunteers.

## 2. Results

The age of the young volunteers (n = 10) was 24.5/20–28 (median/minimum–maximum), the age of the older volunteers (n = 10) was 86.5/80–93 (median/minimum–maximum). In both groups, seven volunteers were female, and three were male.

At concentrations of 0.3 mg/mL, 1.25 mg/mL, 5 mg/mL and 20 mg/mL, GA stimulated phagocytosis of leukocytes in whole blood drawn from young healthy volunteers in a dose-dependent manner (log-linear regression: Pearson’s correlation coefficient r = 0.41, *p* = 0.003). A plateau was not reached (Figure 1). Since the low solubility of GA in aqueous solutions precluded the use of higher concentrations, the highest concentration studied (i.e., 20 mg/mL) was used in all further experiments.

Trypan blue staining of cells revealed that GA at a concentration of 20 mg/mL was not toxic to white blood cells.

Next, the immune stimulating properties of GA on leukocytes from older volunteers in comparison to leukocytes from young volunteers were tested. GA stimulated phagocytosis of *E. coli* by whole-blood leukocytes from both groups, young (Figure 2A) and older volunteers (Figure 2B). Phagocytosis was increased by 120.8% in blood from young and 39.2% in blood from older volunteers (medians of the percent increase in phagocytosis compared to the respective untreated control in each volunteer). CpG as a positive control [11] stimulated phagocytosis of bacteria in young volunteers only (Figure 2A).

As GA stimulated phagocytosis of *E. coli* by blood leukocytes of old and young volunteers, we calculated in a next step the percentage of intracellularly killed *E. coli* by leukocytes of both groups. Intracellular killing of *E. coli* (i.e., the difference of intracellular bacteria after 30 min and 60 min of gentamicin treatment to kill extracellular bacteria) by blood leukocytes of young volunteers was stimulated both by GA (*p* = 0.045) and CpG (*p* = 0.031) (Figure 3A). In the blood of older volunteers, there was a trend towards stronger intracellular killing in GA-stimulated blood over the whole observation period (differences of intracellular bacteria at 30–60 min, 30–120 min, 30–180 min and 30–240 min of gentamicin treatment). This difference was statistically significant at 210 min (*p* = 0.008). CpG was not able to stimulate intracellular killing of *E. coli* in the blood of older volunteers (Figure 3B).

Immune activation by compounds such as GA can be accompanied by a stimulation of the production of pro-inflammatory cytokines. For that reason, concentrations of the cytokines TNF-α and IL-6, which play a significant role in the clearance of a bacterial infection, were measured in the plasma of ex vivo-stimulated whole blood derived from young healthy volunteers after 2 h of stimulation with GA and CpG. TNF-α and IL-6 were slightly higher than the respective concentrations in unstimulated whole blood; the differences, however, failed to reach statistical significance [TNF-α: control 26.7 (7.0/76.5), GA 40.2 (7.0/74.5), CpG 18.0 (7.0/240.1); IL-6: control 20.4 (7.0/305.8), GA 33.9 (10.4/251.9), CpG 18.6 (7.0/646.3), both *p* > 0.05, Friedman test] (Figure 4A,B). In older volunteers, GA, but not CpG, caused a significant increase in TNF-α plasma concentrations (*p* = 0.008). The differences between the IL-6 concentrations did not reach statistical significance (Figure 4C,D).

## 3. Discussion

Immunosenescence leads to a reduced reactivity of the immune system to infectious stimuli. This is one important cause of the increased susceptibility to many bacterial, viral and fungal infections in old age. The influence of age on the adaptive immune response has been addressed in the last decades, e.g., vaccine efficacy decreases in older individuals compared with young adults [12]. Recent interest focusses on the role of the innate immune response as a first line of defense against bacterial invasion and as a modulator of the adaptive immune response. Immunosenescence affects both neutrophil and monocyte function: neutrophils of older volunteers showed reduced chemotaxis in response to GM-CSF, LPS and FMLP stimulation compared to neutrophils from young volunteers [13]. Primary microglial cells and peritoneal macrophages prepared from aged mice phagocytosed less *E. coli* and released less NO and cyto-/chemokines compared to phagocytes from young mice both without activation and after stimulation with agonists of TLR 2, 4 and 9 [14]. Stimulation of monocytes from older persons by Toll-like receptor agonists resulted in a reduced production of proinflammatory cytokines and a higher expression of CX3CR1 [15]. Phagocytosis of opsonized *E. coli* by neutrophilic granulocytes was decreased in older individuals. After pre-stimulation of whole-blood samples from older persons with TNF-α or Toll-like receptor agonists, the oxidative burst of granulocytes in response to formyl peptides was diminished [16]. Here, a TLR9 agonist reliably increased phagocytosis and intracellular killing of *E. coli* by phagocytes in the blood of young volunteers but was unable to stimulate phagocytosis in the blood of old healthy volunteers.

Trained immunity has emerged as a new concept to non-specifically stimulate the immunological memory by epigenetic and metabolic reprogramming in order to increase the resistance of older individuals against infections [12]. However, many compounds effective in young individuals to train innate immunity are not suitable to stimulate phagocytes of older organisms, thereby increasing their infection resistance [14]. This results in a strong demand for immunostimulants, which reliably activate the immune system of the older adult. In addition to vaccination with Bacille Calmette–Guérin (BCG), which has been recently shown to protect older patients from infections [17] but entails a small risk of infection with the attenuated live bacterium itself [18], and tamoxifen [19], this gap may be filled by GA and its components.

The concentrations of GA used in this study are comparatively high for i.v. use: assuming a body weight of 70 kg and a volume of distribution of approx. 0.3 L/kg × 70 kg = 21 L (i.e., the volume of the extracellular space), 21 L × 20 g/L = 420 g of GA would be needed to attain plasma concentrations of 20 mg/mL, i.e., the concentration active in vitro. Even when we assume that the concentrations to elicit a therapeutic effect in vivo are one order of magnitude lower than the concentrations necessary in vitro, 42 g of GA would still be required. Accordingly, active component(s) must be identified to enable the development of a GA preparation suitable for i.v. use. Publications which indicate biological activity of ethanol extracts might be a first step towards that goal [7,20]. Alternatively, a prolonged prophylactic high-dose oral therapy with GA relying on the intestinal absorption of active components showed an extended survival of mice infected with *Plasmodium berghei* [21].

GA preparations at concentrations greater than or equal to those effective in the present study can be used for local applications: mouthwashes of an ethanol extract of GA (62.5 mg/mL H_2_O = 6.25% *w*/*v*) showed promising caries preventive and antibacterial effects with no oral side effects and may substitute chemical agents such as chlorhexidine for safe long-term use [20]. GA has been used in several hydrogel wound dressings to improve their wound healing potential. In lacerations, concentrations of GA greater or equal than those applied in the present study can easily be reached. There, GA can exert both its phagocytosis-stimulating and direct antibacterial properties [22,23]. GA may have a delayed inhibitory effect on bacterial growth after 180 min of co-incubation. In the first 180 min of exposure, bacterial growth may be unchanged or even stimulated as a consequence of the sugars contained in GA.

In conclusion, GA has immunostimulatory properties on leukocytes from both young and older individuals and inhibits growth of some bacteria. Its active component(s) remain to be identified. As a crude aqueous solution or ethanol extract, it may qualify for the topical administration on chronic wounds of aged patients. Its active compound(s) may be suitable for the induction of trained innate immunity in older individuals.

## 4. Materials and Methods

### 4.1. Materials

The natural GA was collected from *Acacia senegal* (L.) Willdenow during summer 2015 from the Kordofan area, West Sudan, with the help of local experienced people. Together with the collected GA, a plant sample from the tree was brought by S.B. to the Herbarium of Sudanese Medicinal and Aromatic Plants & Traditional Medicine Research Institute (MAPTRI), National Center for Research, Khartoum, Sudan [7]. There, the tree, from which GA was collected, was identified and taxonomically authenticated as *Acacia senegal* (L.) Willdenow. It was cleaned from impurities by using a sharp object and then washed with sterile lipopolysaccharide (LPS)-free water three times for 10 min. Then, it was dissolved in LPS-free water at a concentration of 20% (*w*/*v*) and stirred with a magnetic stirrer at room temperature until it was completely dissolved. This solution was termed natural crude aqueous solution of GA [7]. It was used in all experiments at final concentrations of 0.3 mg/mL, 1.25 mg/mL, 5 mg/mL and 20 mg/mL. The oligonucleotide CpG ODN 1668 (CpG class B, 5′ TCC ATG ACG TTC CTG ATG CT, molecular mass 6382.6 g/mol, TIB Molbiol, Berlin, Germany), containing an unmethylated cytosine-guanosin (CpG), a strong Toll-like receptor 9 agonist [11], was used as positive control (concentration 10 µg/mL). 

### 4.2. Volunteers

The study was performed with blood of healthy volunteers belonging to two age groups: (a) 20–35 years (young) or (b) ≥80 years (older adults). Persons with a history of malignant diseases or anti-cancer chemotherapy, a history of autoimmune diseases, an actual medication of a drug with immunomodulatory properties, surgery or a bone fracture within the last 12 months before the study, vaccination within the last 3 months, infection during the last month, a blood hemoglobin ≤11 g/dL or a blood leukocyte count <4000 or >10,000/µL or abnormalities in the differential white blood cell count were excluded. All volunteers ate a well-balanced diet. Persons with a food allergy or intolerance were excluded. Chronic medications without immunosuppressive adverse effects and adequately treated chronic diseases (e.g., hypertension, mild *Diabetes mellitus*) were no contraindication for participation in this study.

### 4.3. Ex Vivo Phagocytosis

Heparinized whole human blood (1 mL), not later than 2 h after drawing, was pre-incubated with GA or CpG oligonucleotides (both dissolved in 0.9% NaCl) or with 0.9% NaCl as untreated control for 2 h. Then, 2.5 × 10^7^ colony-forming units (CFU)/mL *E. coli* K1 were added, and leukocytes of the blood were incubated for 30 min to enable phagocytosis. Thereafter, the tubes containing whole blood, immunostimulants and bacteria were centrifuged at 1500× *g* for 5 min. All supernatants were collected for the measurement of cytokines. Osmotic lysis of the erythrocytes using 0.2% NaCl and 1.6% NaCl for 30 s was performed three times, removing all supernatant fluid after each centrifugation step. Thereafter, the pellet was resuspended in 550 µL of Roswell Park Memorial Institute (RPMI) medium 1640 containing gentamicin (final concentration 100 µg/mL) for 30 min to kill all extracellular bacteria. After further incubations with gentamicin (final concentration 100 µg/mL) at 37 °C for 30 min (young and older volunteers) and for 90, 150 and 210 min (older volunteers) to prevent extracellular bacterial growth and estimate the intracellular bacterial killing rate, 100 µL of the samples were centrifuged, washed once in 0.9% NaCl and then centrifuged again. The final pellets were resuspended in 100 µL of bi-distilled water to lyse the eukaryotic cells. Bacterial concentrations in the lysates and the suspensions were determined by 1:10 quantitative plating on agar plates containing 5% sheep blood. Bacterial killing in a given time interval (30 min in the blood of young and older volunteers, 30, 90, 150 and 210 min in the blood of older volunteers because of the slower bactericidal activity) was defined as the concentration measured at 30 min minus the concentration measured at 60 (young and older volunteers), and 90, 150 or 210 min (older volunteers). 

### 4.4. Cytokines

Cytokines were measured in the supernatants after 2.5 h of GA or CpG stimulation by enzyme immunoassay (ELISA MAX Standard Set Human IL-6 and Human TNF-α, BioLegend^TM^, San Diego, CA, USA) according to the manufacturer’s instructions.

### 4.5. Cell Viability 

To determine whether GA has a cytotoxic effect on blood cells, staining with trypan blue was performed after 2.5 h incubation with either GA or 0.9% NaCl as control. Briefly, after incubation with GA, cells were washed and erythrocytes lysed firstly by osmotic lysis using 0.2% NaCl and 1.6% NaCl and secondly using a FACS lysing solution (BD Biosciences, Heidelberg, Germany). Afterwards, the cells were resuspended in either phosphate-buffered saline (PBS) or 4% paraformaldehyde (PFA) and stained with trypan blue.

### 4.6. Statistics

Blood of each volunteer was studied in triplicates, and the medians of the measurements were used for statistical analysis performed by GraphPad Prism 6 Software (GraphPad, San Diego, CA, USA). The dose–response relation was established by log-linear regression, and the calculation of the Pearson’s correlation coefficient was performed. Comparisons were performed by the nonparametric Friedman test, followed by a Wilcoxon matched-pairs signed-rank test with Bonferroni–Holm correction for repeated testing. *p* values *p* ≤ 0.05 were considered statistically significant.

## 5. Conclusions

GA has immunostimulatory properties on leukocytes from both young and older individuals and inhibits growth of some bacteria. Its active component(s) remain to be identified. As a crude aqueous solution or ethanol extract, it may qualify for the topical administration on chronic wounds of aged patients. Its active compound(s) may be suitable for the induction of trained innate immunity in older individuals.

## Figures and Tables

**Figure 1 antibiotics-13-00482-f001:**
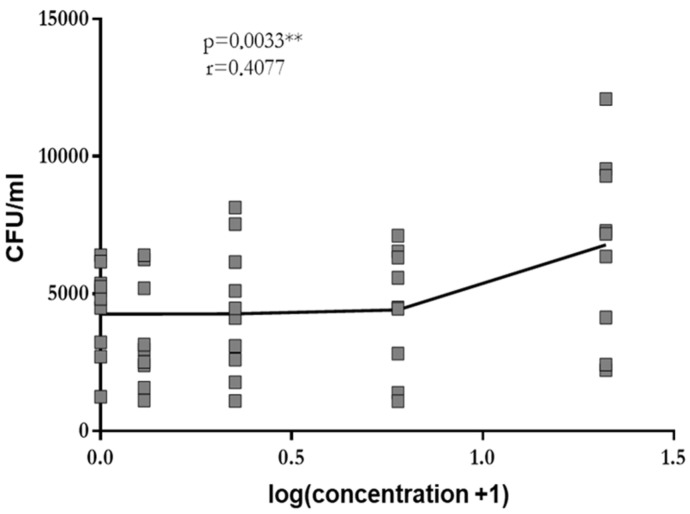
Phagocytosis of *E. coli* by leukocytes in whole blood drawn from young volunteers after 2 h of pre-incubation with 0.0 mg/mL, 0.3 mg/mL, 1.25 mg/mL, 5 mg/mL and 20 mg/mL of GA. Each point represents the median of 3 measurements. The dose–response relation was established by log-linear regression, and the calculation of the Pearson’s correlation coefficient was performed. ** *p* < 0.01.

**Figure 2 antibiotics-13-00482-f002:**
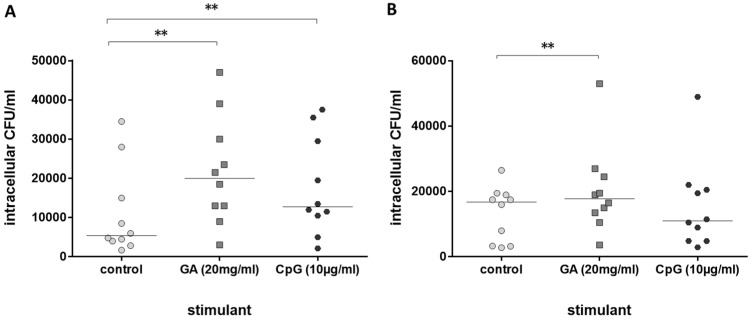
Phagocytosis of *E. coli* by leukocytes in whole blood after 2 h of pre-incubation with 20 mg/mL of GA, 0.9% NaCl as negative and 10 µg/mL of CpG as positive control ((**A**): n = 10, young healthy volunteers, 20–35 years; (**B**): n = 10, older healthy volunteers ≥80 years). Each point represents the median of 3 measurements, the horizontal bars represent the medians of 10 individuals; ** *p* ≤ 0.01, Friedman test followed by Wilcoxon matched-pairs signed-rank test with Bonferroni–Holm correction for repeated testing.

**Figure 3 antibiotics-13-00482-f003:**
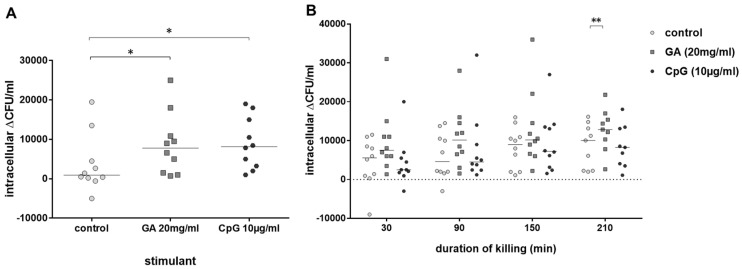
Intracellular killing of *E. coli*. Intracellular killing during 30 min (difference of the intracellular bacterial concentrations after 30 and 60 min of gentamicin treatment to kill extracellular *E. coli*) after previous incubation with 20 mg/mL of GA, 0.9% NaCl as negative and 10 µg/mL of CpG as positive control (n = 10, young volunteers, 20–35 years) (**A**). Killing of *E. coli* in whole blood during 30–210 min (difference of the intracellular bacterial concentrations at 30–60 min, 30–120 min, 30–180 min and 30–240 min of gentamicin treatment) after addition of gentamicin (n = 10, older volunteers, ≥80 years) (**B**). Each point represents the median of 3 measurements in one individual volunteer, the horizontal bars represent the medians. * *p* ≤ 0.05, ** *p* ≤ 0.01, Friedman test followed by Wilcoxon matched-pairs signed-rank test with Bonferroni–Holm correction for repeated testing.

**Figure 4 antibiotics-13-00482-f004:**
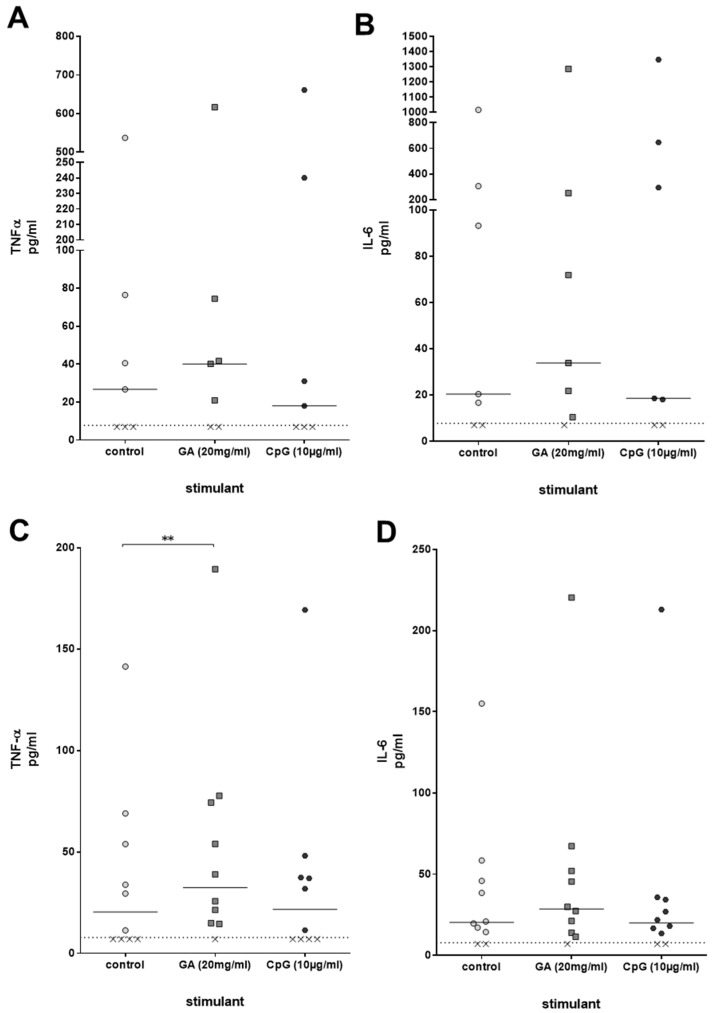
TNF-α (**A**,**C**) and IL-6 (**B**,**D**) concentrations [pg/mL] were measured in plasma of whole blood donated from young (n = 7, 20–35 years) (**A**,**B**) and older (n = 10, age ≥ 80 years) (**C**,**D**) volunteers, which was stimulated with GA or CpG ex vivo for 2 h. Each point represents the concentration measured in one individual, the horizontal bars represent the medians. The detection limit (dotted line) of the enzyme immunoassays was 7.8 pg/mL. Data were analyzed by Friedman test followed by Wilcoxon matched-pairs signed-rank test with Bonferroni–Holm correction for repeated testing. x value = sample below detection limit. ** *p* ≤ 0.01.

## Data Availability

Anonymized primary data are available from the corresponding authors upon reasonable request.

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
