# Peer review of "Gum Arabic Increases Phagocytosis of Escherichia coli by Blood Leukocytes of Young and Old Healthy Volunteers"

_antibiotics, 2024, doi:10.3390/antibiotics13060482_

Round 1
Reviewer 1 Report
Comments and Suggestions for Authors
Latin names must be written in italics, check the title and also check all parts of the manuscript.
Delete the "6. Patents" section, it is not necessary for this manuscript.
in the section "236 4.2. Volunteers" there must be more detail about the characteristics of the subject, diet or eating patterns, inclusion and exclusion criteria and the most important thing is the ethical approval number.
Author Response
- Latin names are now written in italics.
- As requested, we deleted the "Patents" section.
- In the section "Volunteers" we added the folowing sentences: "All volunteers ate a well-balanced diet. Persons with a food allergy or intolerance were excluded. Chronic medications without immunosuppressive adverse effects and adequately treated chronic diseases (e.g. hypertension, mild Diabetes mellitus) were no contraindication for the participation in this study."
- The ethical approval number (3/5/15) is now included in the Institutional Review Board Statement.
Reviewer 2 Report
Comments and Suggestions for Authors
The manuscript: Gum arabic increases phagocytosis of Escherichia coli by blood 2 leukocytes of young and old healthy volunteers.
Present important date about Gum Arabic, a polysaccharide exudate from Acacia senegal (L.) Willdenow trees, has 15 already been used by African native people in natural medicine.
This study aimed at assessing the immunomodulatory effect of GA on phagocytosis and intracellular killing of E. coli by blood leukocytes of young in comparison to older volunteers. 1. In the manuscript did not comment the authorization from Committee, only demonstrated assessing of young comparison to older volunteers.
2. I did not use the plagiarism program.
3. The manuscript present references related with the study.
4. The manuscript is accepted for publication.
Author Response
The sections Institutional Review Board Statement and Informed Consent Statement already contain the information requested by this reviewer:
"Institutional Review Board Statement: The study was conducted in accordance with the Declaration of Helsinki and was approved by the Ethics Committee of the University Medicine Göttingen, Germany (approval number 3/5/15)."
"Informed Consent Statement: Written informed consent was obtained from all subjects involved in the study."
We do not want to duplicate this information. If you prefer to have it both in the Materials & Methods section and the Institutional Review Board and Informed Consent Statements, we suggest to include the following sentences into the Methods section: "The study was approved by the Ethics Committee of the University Medicine Göttingen, Germany (approval number 3/5/15). Written informed consent was obtained from all subjects participating in this study."
Reviewer 3 Report
Comments and Suggestions for Authors
A useful study with interesting results that could be used further in different fields. However, I am pointing out some important parts that need to be clarified for the study to be relevant.
1. First of all, the way the authors described the preparation of the extract by dissolving the gum in water - is not actually an extract, but a solution. Extract preparation includes different techniques such as maceration or Soxhlet extraction and does not involve complete dissolution of the herbal drug (in this case the Arabic gum). Instead, the obtained liquid extract need to be filtered and usually (not necessary) evaporated and then the mother solution should be made and further dissolved appropriately. The main text should be adjusted accordingly.
2. The first subsection in the Material and Methods section (lines 224-235) should be rewritten – it is not clear how the authors collected the Arabic gum, I assume it was collected in the Kordofan region, but in that case, it should be clearly given the locality with coordinates. In that sense, the authors did not obtain the extract (as it is declared in the very beginning of the first sentence) but the gum as an herbal drug from which they prepared the extract – more precisely the solution. In the following sentence the authors declared who did and where was done the taxonomic identification of the biological source (S. senegal plant). However, if they really obtained the extract but not the gum (the drug), as it is declared in the very first sentence, then it is not clear how the authors did the taxonomic identification of the biological source (the plant material) – since they did not have the plant material. If they performed macroscopic or microscopic identification of the gum instead, then the taxonomic identification of the biological source (S. senegal plant) was not done, because it is not the appropriate way to taxonomically identify any plant species (instead, the authors identified – not taxonomically – the herbal drug, which is, in this particular case, the Arabic gum).
Therefore, please pay attention to all of that and clarify what the authors actually did and if the extract preparation should be also rewritten since it might be done previously (as the authors declared the extract was obtained); in that case, a place (institution/laboratory) and extract preparation should be highlighted and again – make a difference between an extract and a solution of the gum used.
3. Everywhere should be used appropriate Latin names, and this is particularly addressed to the plant species (Acacia senegal) used as a biological source for the Arabic gum which is, according to the most contemporary taxonomic principles actually a synonym of Senegalia senegal (L.) Britton. Please, change that throughout the whole manuscript and pay attention to the second word in the plant name – it is always written with the small capital letter (the authors used abbreviation A. Senegal, which is also not an option, but A. senegal – in this case it would be S. senegal; I guess it typewriting but, in any case, it should be corrected).
4. Naming bacterial species by the full Latin names in the first mentioning should not be along with the genera abbreviation: e.g. line 42 when the authors wrote ‘’Escherichia (E.) coli’’. Simply use an abbreviated form after the first mention in the main text.
5. The Results section should not contain the aim of the study (lines 82-83).
Author Response
- We clarified that we used an aqueous solution of Gum Arabic. In the Introduction and in the Discussion we clearly stated, when data were generated with a solution and when they were generated with an ethanol extract.
- As already written, the Gum Arabic was collected in the Kordofan region. We are unable to give the coordinates. We attach the Certificate from Khartoum to this reply (Gum Arabic Certificate Sudan.pdf). We clarified the first subsection of Materials and Methods as follows: "The natural GA was collected from Acacia Senegal during summer 2015 from the Kordofan area, West Sudan, with the help of local experienced people. Together with the collected GA, a plant sample from the tree was brought by S.B. to the Herbarium of Sudanese Medicinal and Aromatic Plants & Traditional Medicine Research Institute (MAPTRI), National Center for Research, Khartoum, Sudan [7]. There, the tree, from which GA was collected, was identified and taxonomically authenticated as Acacia senegal (L.) Willdenow."
Acacia senegal (L.) Willdenow is a synonym of Senegalia senegal (L.) Britton. This was stated in the Introduction. Since the MAPTRI authenticated Acacia senegal (L.) Willdenow, we used this term throughout the text. - In the revision, we use appropriate Latin names throughout the text.
- In line 42 we deleted "(E.)".
- As requested, we deleted the first sentence of the Results section: "This study aimed at assessing the immunomodulatory effect of GA on phagocytosis and intracellular killing of E. coli by blood leukocytes of young in comparison to older volunteers."
We thank all reviewers and in particular this reviewer for the very helpful comments to improve the manuscript!
